# Hypoxic State of Cells and Immunosenescence: A Focus on the Role of the HIF Signaling Pathway

**DOI:** 10.3390/biomedicines11082163

**Published:** 2023-08-01

**Authors:** Dario Troise, Barbara Infante, Silvia Mercuri, Giuseppe Stefano Netti, Elena Ranieri, Loreto Gesualdo, Giovanni Stallone, Paola Pontrelli

**Affiliations:** 1Nephrology, Dialysis and Transplantation Unit, Advanced Research Center on Kidney Aging (A.R.K.A.), Department of Medical and Surgical Science, University of Foggia, 71122 Foggia, Italy; dario.troise@unifg.it (D.T.); barbarinf@libero.it (B.I.); silvia.mercuri1990@gmail.com (S.M.); giovanni.stallone@unifg.it (G.S.); 2Clinical Pathology Unit, Department of Medical and Surgical Sciences, University of Foggia, 71122 Foggia, Italy; giuseppestefano.netti@unifg.it (G.S.N.); elena.ranieri@unifg.it (E.R.); 3Nephrology, Dialysis and Transplantation Unit, Department of Precision and Regenerative Medicine and Ionian Area (DiMePRe-J), University of Bari “Aldo Moro”, Policlinico, Piazza Giulio Cesare 11, 70124 Bari, Italy; loreto.gesualdo@uniba.it

**Keywords:** hypoxia-inducible factors, cellular aging, cellular energy, immunosenescence

## Abstract

Hypoxia activates hypoxia-related signaling pathways controlled by hypoxia-inducible factors (HIFs). HIFs represent a quick and effective detection system involved in the cellular response to insufficient oxygen concentration. Activation of HIF signaling pathways is involved in improving the oxygen supply, promoting cell survival through anaerobic ATP generation, and adapting energy metabolism to meet cell demands. Hypoxia can also contribute to the development of the aging process, leading to aging-related degenerative diseases; among these, the aging of the immune system under hypoxic conditions can play a role in many different immune-mediated diseases. Thus, in this review we aim to discuss the role of HIF signaling pathways following cellular hypoxia and their effects on the mechanisms driving immune system senescence.

## 1. Introduction

Aging is a physiological time-related process affecting the majority of living organisms and is characterized by cellular and functional deterioration. It results in an increased susceptibility to the development of many diseases, including cancer, as well as cardiovascular, metabolic, and neurodegenerative diseases [1]. Aging can be considered a mosaic of interactions between abnormal molecular signaling pathways, in which hypoxia plays an important role. Despite atmospheric oxygen levels being nearly 20%, its availability in human organisms is variable, and oxygen is distributed from the respiratory system to every tissue by blood flow; thus, not all cells are exposed to the same oxygen concentration [2]. Low oxygen concentration is associated with senescence, and this condition is characterized by the activation of signaling pathways and transcription factors, termed hypoxia-inducible transcription factors (HIFs), that regulate the expression of several genes involved in aging [3].

HIFs can also influence several aspects of immune cells, from viability to biological function and differentiation. Immune cells are susceptible to hypoxic conditions, and HIFs have a pivotal role in maintaining cellular homeostasis and in supporting high energy-dependent processes in the presence of low O_2_ concentrations. All immune cells express HIF-1, whereas neutrophils, activated T-cells, and NK cells selectively express HIF-2 [4].

This review aims to emphasize the role of HIF signaling pathways following circumstances that cause cellular hypoxia and its role in immunosenescence.

## 2. HIF Structure and Biology

HIFs are heterodimeric transcription factors, which include three distinct members (HIF-1, HIF-2, HIF-3) formed by an α subunit (O_2_-sensitive subunit) and a β subunit (O_2_-insensitive subunit) [5]. Among them, HIF-1 plays a key role in regulating cellular response to hypoxia. Semenza et al., observed that expression of the erythropoietin gene in human liver and kidney is inducible by anemia or hypoxia [6]. Moreover, Kulkami et al., discovered that a single DNA sequence (5′-RCGTG-3′) located in the 3′-flanking region of the erythropoietin (EPO) gene, known as hypoxia response elements (HREs), plays a pivotal role in the transcription of genes under hypoxic conditions [7]. A decrease in oxygen concentration leads to the production of HIF-1, which recruits co-activator proteins to form active transcriptional complexes and bind to HRE [8].

Hypoxia-inducible factor-1α (HIF-1α) and hypoxia-inducible factor-1β (HIF-1β or ARNT) dimerize to form HIF-1 [9]. Moreover, HIF-1α contains two trans activating domains termed CTAD (carboxyl-terminal domain) and NTAD (amino-terminal domain); both are crucial for optimal HIF transcriptional activity. Whereas CTAD is necessary for full HIF activity, NTAD provides target gene specificity [10]. HIF-1 interacts with HRE to regulate more than 100 genes [11]. Whereas HIF-1α is upregulated in low oxygen conditions, HIF-1β is expressed constitutively [12].

In humans, three HIF-α subunits (HIF-1α, HIF-2α, HIF-3α) are known, and they dimerize with the same β subunit (HIF-1β). They are products of different genes: hypoxia-inducible factor 1 subunit α (HIF-1A, HIF-1α), endothelial PAS domain-containing protein 1 (EPAS-1, HIF-2A, HIF-2α), and hypoxia-inducible factor 3 subunit α (HIF-3A, HIF-3α), whose expression is protected under low O_2_ concentrations [13]. Although HIF-1α mediates the majority of adaptive changes in response to hypoxia conditions and is best studied, HIF-2 is connected to several signaling pathways, such as tumor invasion, angiogenesis, cell migration, and lipid metabolism regulation [14]. HIF-1α also mediates the expression of genes involved in lipid metabolism, such as FASN (fatty acid synthase), an essential lipogenic enzyme that promotes adequate fatty acid supply. AGPAT2 (acylglycerol-3-phosphate acyltransferase 2) and LPIN1 (lipin-1), two enzymes of the triacylglycerol biosynthetic pathway, have been demonstrated to mediate lipid droplet accumulation as a result of exposure to hypoxia. Moreover, hypoxia promotes lipogenesis through the induction of genes involved in the absorption, synthesis, and storage of fatty acids in an HIF-dependent manner [15]. HIF-3α, a much less studied isoform, is subjected to complex alternative splicing. It is a negative regulator with the capacity to inhibit the HIF pathway binding to ARNT, but it also seems to possess transactivation capacity, depending on the HIF-3α isoform. Tolonen et al., demonstrated that, when co-overexpressed with HIF-1α or HIF-2α, HIF-3α2 (a splicing variant of HIF-3α) is more likely to have synergistic effects than inhibitory effects, inducing the expression of a subset of genes, including erythropoietin (EPO) [16].

In normoxic conditions, HIF-1α is rapidly degraded by the von Hippel-Lindau tumor suppressor protein (VHL); therefore, it has a short half-life (5 min).

Its degradation occurs because of a recognition of the HIF-1α protein by prolyl hydroxylase proteins (PHDs), which serves as an oxygen sensitivity system for the HIF pathway before ubiquitination. PHDs use oxygen as a substrate to cause hydroxylation of HIF-1α proline residues (PRO402 and PRO564). The VHL protein mediates degradation via the ubiquitin proteasome pathway via association with elongins B and C, cullin 2, and ring-box 1 to form an E3 ubiquitin ligase complex that cause ubiquitination of HIF-1α [17].

When the oxygen concentration drops below dangerous levels, PHDs are inactivated and HIF-1α and HIF-1β can translocate to the nucleus, dimerize, and bind to p300/CBP to form a transcriptional activation complex that binds to HRE and activates the transcription of target genes [18] (Figure 1).

In addition, another regulator of the transcriptional activity of HIFs is a member of the Fe^2+^ and α-ketoglutarate-dependent dioxygenase family, termed FIH-1 (factor inhibiting HIF-1) or HIF asparaginyl hydroxylase. FIH-1 suppresses HIF-1’s transactivation function and prevents the association of HIF-1α with the p300 coactivator protein via the hydroxylation of an asparaginyl residue (ASN803), utilizing O_2_ and ketoglutarate as substrates. In the absence of the FIH-1 isoform, the Asn803 residue is not hydroxylated [19]. Disinhibition of HIF-1α C-terminal domain activity is sufficient for repression of FIH-1 activity during hypoxia. The HIF-1α isoform is more prone to FIH-1 alteration than the HIF-2α, which is the second main HIF isoform, and since many of the enzymes that control metabolism are directly regulated by the HIF-1α isoform, a differential control of HIF-1a by FIH may be able to cause specific changes in metabolic response. FIH-1 is an active hydroxylase at oxygen concentrations lower than those necessary to inhibit the activity of PHD because of its higher oxygen affinity than PHD enzymes. In these situations, when a rapid onset of hypoxia outpaces the process of HIF-1α accumulation due to negative PHD regulation, FIH-1 might have a protective role when the cell faces a metabolic disadvantage. Sim et al., showed that loss of FIH activity accelerated oxidative reactions, which lowered intracellular oxygen levels and potentiated PHD inhibition, leading to the accumulation of HIF-1α. Ultimately FIH-1 and the PHD/VHL pathway acted in cooperation to accelerate HIF-mediated metabolic responses to hypoxia [20].

## 3. Reduced Oxygen Supply and Cellular Energy Status

It is useful to correlate cellular adaptation, caused by reduced oxygen supply, to cellular energy status, since reduced perfusion of tissues is often coupled with lack of nutrients. The metabolism of cells exposed to hypoxia is reprogrammed by HIF-1, and this affects mitochondrial biogenesis, mitophagy, and dynamics. Moreover, HIF-1 regulates the efficacy of the respiratory chain, promoting adaptation to hypoxic conditions by reducing cellular oxygen dependence and therefore reducing mitochondrial activity [21].

In normoxia, ATP generation occurs through oxidative phosphorylation based on the oxidation of acetyl CoA, leading to a reduction in NAD+ and FAD+ in NADH and FADH, donors of electrons to the mitochondrial electron transport chain (ETC). A proton gradient is formed, and it is used to drive the formation of ATP. In hypoxia, HIF-1 increases the uptake of glucose to promote glycolysis and the transcription of SLC2A1 (solute carrier family 2 member 1) and SLC2A3 (solute carrier family 2 member 3) genes, which encode the glucose transporters GLUT1 and GLUT3. Debangshu et al., showed how the immune response is influenced by the effects of HIF-1 in hypoxic cells. Tumor cells induce glucose deprivation for T cells, resulting in a state of T-cell anergy and death because of their dependence on glycolysis [22]. The accumulation of H+ and lactate is the result of the transcription of numerous genes involved in maintaining host homeostasis, such as lactate dehydrogenase A (LDHA), hexokinases (HK1 and HK2), enolase (ENO1), phosphoglycerate kinase (PGK1), and pyruvate kinase; all of these are the result of metabolic reprogramming towards glycolysis mediated by HIF-1 and aimed at the production of ATP in hypoxic cells [23].

ETC activity is influenced by hypoxia on various levels. Mammalian cytochrome c oxidase (COX) is the terminal complex (complex IV) of ETC. It can function at very low O_2_ levels because of a high affinity between O_2_ and COX, allowing cells to largely maintain their ATP levels [24]. COX is formed by 13 subunits: 3 catalytic subunits and 10 regulatory nuclear-encoded subunits. HIF-1 induces the expression of different COX isoforms to regulate respiration, which induces the COX4 isoform 2 (COX4I2) subunit and the expression of the mitochondrial protease LON for proteasomal degradation of COX4 isoform 1 (COX4I1), allowing for the incorporation of the COX4I2 subunit into COX and a more efficient transfer of electrons to O_2_ under acute onset hypoxia [25].

Contrarily, in order to decrease the generation of reactive oxygen species (ROS) under prolonged hypoxia, the activity of the I, II, and III ETC complexes is reduced. ETC activity is reduced by the induction of the mitochondrial NDUFA4L2 (NADH dehydrogenase (ubiquinone) 1 alpha subcomplex, 4-like 2) gene in a HIF-1-dependent manner, which decreases complex I activity and respiration. Moreover, several microRNAs (miRNAs) are induced through HIF-1, including mir-210, which represses iron–sulfur cluster assembly proteins 1 and 2 (ISCU1/2), two components that are necessary for the correct assembly of iron–sulfur (Fe-S) clusters, which are essential for electron transport and mitochondrial oxidation–reduction processes within ETC complexes I, II, and III [26]. Furthermore, miR-210-mediated downregulation of ETC components, such as subunit D of succinate dehydrogenase complex (SDHD), a complex II subunit; and NADH dehydrogenase (ubiquinone) 1 alpha subcomplex, 4, 9 kDa (NDUFA4), a complex I subunit, could participate in mitochondrial dysfunction associated with modulation of HIF-1 activity [27].

In addition, the hypoxia-inducible gene domain family member 1A (HIGD1A) is another HIF-1-dependent protein that raises COX activity, maintaining mitochondrial integrity and improving cell viability in hypoxic environments, as described by Jia YZ et al. [28].

Another protein that has a key role in sensing the impairment of cellular energy status is AMP-activated protein kinase (AMPK), which promotes ATP generation through the activation of catabolic pathways and prevents ATP depletion through the inhibition of anabolic pathways. It is a very old and well conserved mechanism in eukaryotic cells and is implicated in hypoxic cellular adaptation [29].

Low oxygen concentration is often associated with the activation of AMPK pathways mainly via the LKB1 (liver kinase B1 or STK11)-AMPK axis because of the decreased ATP/AMP ratio. Actually, oxidative phosphorylation dysfunction in mitochondria does not just result in the production of reactive oxygen species that directly activate AMPK but also activates the AMP-protein kinase through serine threonine kinase LKB1, a known tumor suppressor gene. Although it has been suggested that AMPK and HIF interact to protect cells from hypoxia, additional research must be conducted on this topic [30]. There is scientific evidence that LKB1 somatic-inactivated mutations play a role in the development of many types of cancers. LKB1 is involved in the modulation of several metabolic pathways and modulates glucose metabolism through glycolysis, aerobic oxidation, the pentose phosphate pathway, and gluconeogenesis. It controls several other transduction pathways as well, such as lipid, glutamine, and serine metabolisms. In LKB1- and AMPK-deficient cells, HIF-1α is upregulated to provide energy and preserve the cellular shape of LKB1-AMPK-deficient cells (e.g., some types of cancer cells) [31].

Regarding mitochondrial dynamics, the LKB1-AMPK pathway is known to restore the function of damage-activating fusion processes but the upregulation of HIF-1α in the tumor mitochondria increases the activity of dynamin-related Protein 1 (DRP1) and unbalances mitochondrial dynamics toward fission by downregulating the expression of mitofusin-1 (MFN1) and optic atrophy 1 (OPA1) [32].

AMPK inhibits mammalian target of rapamycin (mTOR), which modulates protein translation at both the initiation and the elongation steps. These effects lead to a reduction in the translation of nuclear-encoded mitochondrial transcript through 4E-BP1, also known as eukaryotic translation initiation factor 4E-binding protein 1, which is implicated in increasing the capacity of ATP synthesis for cell growth [33].

Hypoxic insult then leads to a change in protein synthesis; the protein translation is downregulated in order to suppress all the cell expensive energy processes and to prevent cellular stress induced by the abnormal accumulation of unfolded and/or misfolded proteins at the endoplasmic reticulum (ER). These processes occur mainly at the translation initiation level and are driven by two pathways: downstream of PERK (protein kinase R-like ER kinase) and the mTOR complex [34].

Eukaryotic cells have developed a survival mechanism to decrease ER stress; specifically, they trigger the unfolded protein response (UPR), which is involved in the degradation of misfolded proteins. In this way, they increase the production of chaperones implicated in protein folding and cellular apoptosis if that control system fails [35]. PERK is considered a UPR sensor that has a protective role during hypoxic cell stress. It phosphorylates eukaryotic initiation factor 2 alpha (eIF2α) and inhibits translation initiation. Moreover, mTOR regulates cap-dependent translation through the phosphorylation of eIF4E-binding proteins (4E-BPs). The 4EBPs normally prevent assimilation of eIF4E into the eIF4F complex (formed by eIF4A, eIF4G, and eIF4E), which is necessary for binding to the mRNA cap. Hypoxia seems to have a negative effect on the regulation of mTOR; therefore, it is not able to phosphorylate 4EBPs and initiate cap-dependent translation, resulting in a decrease in protein synthesis [36].

HIF-α mRNA translation is promoted by upregulation of the PI3K/AKT/mTOR pathway [37]. Land et al., demonstrated that activation of mTOR enhances the activity of HIF-α during hypoxia and that treatment with mTOR-inhibitors can reduce the high levels of HIF activity in cells [38]. Under hypoxic environments, mTOR signaling is negatively regulated by HIF-1 target gene DNA damage inducible transcript 4 (DDIT4/REDD1). DDIT4/REDD1 inhibits the activity of Rheb and inactivates mTOR in order to enhance the autophagy process [39], which has a pivotal role in recycling mitochondria and other organelles, the endoplasmic reticulum, and peroxisomes [40].

With the aging process, cells become more and more susceptible to ischemic damage, and the incidence of ischemia increases with age [41]. In this context, aging is linked to a significant risk of ischemia/reperfusion (IR) injury, characterized by a reduction in blood flow to the tissue (ischemia), which is then restored (reperfusion), causing damage to the ischemic tissue [42]. Many organs and tissues are affected by IR injury, including heart [43], kidney [44], lung [45], brain [46], and liver [47], causing increased mortality and morbidity worldwide. Numerous mechanisms on how ischemia/reperfusion injury induce cell death have been described. One of these is apoptosis. Hypoxic insult induces both intrinsic and extrinsic pathways through the activation of a caspase cascade, which leads to proteolytic cell death. Moreover, IR injury can trigger mitoptosis, necroptosis, and autophagy [48]. In particular, autophagy is a process characterized by the formation of double-membrane vesicles, termed phagosomes, that contain cytosolic proteins and organelles and go through a process called phagosome maturation, which culminates in the fusion of phagosomes with lysosomes catabolized by lysosomal degradative enzymes. Autophagy prevents the biogenesis of lipids and proteins and optimizes the use of limited energy supplies. This process has a pivotal role in cellular response to stress and does not appear to function well when the cell goes through the senescence process [49]. Severe hypoxic conditions can induce autophagy through several mechanisms in which HIF may or may not be involved. Regarding HIF-dependent mechanisms, HIF-1α-dependent expression of BNIP3 (Bcl-2/adenovirus E1B 19-kDa-interacting protein) causes the activation of signal path-induced autophagy during low O_2_ availability. Additionally, there is a severe nutritional depletion that frequently occurs in hypoxic environments that can lead to autophagy through HIF-independent mechanisms such as AMPK through Unc-51-like kinase (ULK-1) complex and PI3K [50] (Figure 2).

Tissues that undergo ischemic insult or reperfusion benefit from the transcriptional response of HIF. Its transcription factors exert a protective influence on ischemic/reperfused tissues because of the rapid restoration of ATP generation capacity. Moreover, HIF can increase the production of angiogenic factors, allowing for the restoration of blood supply to tissues [51]. Hong et al., demonstrated the effect of cilastatin preconditioning in renal IR damage by enhancing hypoxia inducible factor pathways and decreasing HIF-1α ubiquitination [52]. Ischemic preconditioning (IPC) can be considered a protective intervention to limit IR injury. It is defined as an exposure to a stimulus to protect tissues from IR damage, and HIF-1α represents an attractive target in this process [53]. According to Cai et al., who examined the role of HIF-1α in IPC, cardiomyocyte cell death was reduced in mice expressing the HIF-1α gene compared to wild-type mice exposed to IPC, resulting in improved heart function [54]. Furthermore, Jia et al., demonstrated upregulation of miRNA-21 consequently to IPC and activation of HIF-1α, reducing the generation of proinflammatory cytokines in target organs [55]. Evidence shows that the expression of HIF-1α protects cells from damage and allows them to adapt to hypoxic responses during myocardial ischemia/reperfusion injury [56]. IPC also exerts a protective effect against hypoxic/ischemic brain injury through the upregulation of HIF-1α, which is involved in the activation of autophagy pathways, as shown by Lu et al. [57]

## 4. HIFs and Immunosenescence

Immunosenescence is characterized by an extensive modification in the innate and adaptive response that leads to decreased immune response in the elderly, involving dysregulation in T-cell response [58], deprived B-lymphopoiesis [59], and impaired activity of antigen-presenting cells [60]. Aging is a significant risk factor for cancer, and the immune system plays a crucial immune surveillance role in the antitumor response but is also directly related to the development and progression of neoplasms. In this context, immunosenescence plays a key role in cancer risk, since the tumor microenvironment (TME) may affect various aspects of the immune system, accelerating its aging [61].

Programmed death-ligand 1 (PD-L1), an immunoinhibitory molecule, is encoded by the Pdcdl1 gene on chromosome 9 and is expressed by several immune cells within the TME. It binds programmed death-1 (PD-1), a T-cell co-inhibitory receptor, to promote immunosuppression, resulting in high tumor aggression and evasion of cancer cells from immune response [62]. Hypoxia is frequently present in the TME in solid tumors; it induces HIF-driven transcriptional responses in cancer cells, where a direct link between PDL-1 and HIF-1α has been reported [63].

Once the immune system is activated, ATP demand in immune cells increases and a metabolic shift is required to support cell proliferation and activity. Since immune cells are not able to use oxidative metabolism to support metabolic needs in hypoxic conditions, glycolysis, an oxygen-independent pathway, is required to supply ATP [64]. The first evidence of HIFs’ role in the control of the immune system was reported in 2003 by Cramer et al. [65]. Till then, several studies focused on the role of HIFs in the regulation of innate and adaptive immune cells, such as T and B cells, NK cells, dendritic cells, neutrophils, macrophages, and epithelial cells [66,67,68].

Innate immune system cells, such as neutrophils, dendritic cells, and macrophages, undergo phenotypic and metabolic changes in response to low oxygen environment, and activation of HIF precedes these adaptive processes. HIF-1 and HIF-2 affect several functions of macrophages, such as cytokine production, motility, and response to viruses and bacteria [69,70]. Depending on the stimuli, macrophages can polarize into the pro-inflammatory phenotype, which predominantly uses glycolysis as a source of ATP (M1-type macrophages), and the anti-inflammatory phenotype, which uses oxidative phosphorylation (M2-type macrophages) for ATP generation [71]. High expression of HIF-1α induces M1-type macrophages [72], which, through glycolysis, leads to accumulation of Krebs cycle intermediate. In particular, high levels of succinate can inhibit PDHs and therefore lead to the stabilization and accumulation of HIF-1α [73]. Moreover, upregulation of MTORC1 is associated with M1 polarization [74], and a reduction in pro-inflammatory macrophages expression has been observed through the inhibition of HIF-1α and mTOR [75]. Hypoxia can also induce gene expression in human and murine macrophages in a HIF-independent way by upregulating NF-kβ (nuclear factor kappa-light-chain-enhancer of activated B cells), activating transcription factor 4 (ATF4) and early growth response-1 [76]. Fangradt et al., in fact, demonstrated the presence of HIF-1 in the cytoplasm of primary human monocytes and macrophages, whereas NF-kB1 (p50) was found in monocyte nuclei under hypoxic conditions [77].

Dendritic cells (DC) can be considered a key link between innate and adaptive immune systems; thus, it is important to evaluate the role played by hypoxia on DC and its influence on the quality and intensity of immunoreaction. Jantsch et al., demonstrated that hypoxia alone cannot activate murine dendritic cells, but hypoxia combined with lipopolysaccharides (LPS) can induce HIF-dependent changes affecting cell survival, migration, and T-cell activation [78]. Nandini et al., demonstrated that hypoxia exposure in human monocyte-derived dendritic cells is associated with upregulation of HIF-1α and downregulation of anti-apoptotic molecule Bcl-2, thus resulting in cellular death. Cell death was not observed in LPS-induced mature DC, despite high expression of HIF-1α via PI3K/Akt pathway; however, the inhibition of this signaling pathway was correlated with the death of hypoxic mature DC [79].

Upregulation of glycolysis is associated with activation of DCs via toll-like receptors (TLRs). The glycolytic process in DC can be divided in two phases: acute induction, which occurs within minutes, lasts for hours, and is necessary for early DC maturation; and long-term induction, which is mandatory for metabolic adaptation. HIF-1α activity depends on mTOR and promotes inducible nitric oxide synthase (iNOS) expression during long-term induction, leading to a NO-mediated suppression of DC mitochondrial activity. It has been demonstrated that mTOR inhibition is ineffective in controlling DCs’ acute induction of glycolysis, but it is required for the reduction of NO production, which leads to increased DC mitochondrial activity [80]. Although the role of HIF-1α in DCs needs to be more precisely defined, there is evidence supporting its involvement in the activation, maturation, and migration of dendritic cells with a pro-inflammatory profile [81].

Hypoxic conditions induce a metabolic switch towards glycolysis and in polymorphonuclear neutrophils (PMNs), as described in macrophages and dendritic cells. PMNs are cells of the innate cellular system. After migration to sites of infection and inflammation, they produce ROS and proinflammatory cytokines like TNF-α, IL-1β, and interferons [82]. Moreover, they can entrap and kill pathogens through neutrophil extracellular traps (NETs) [83]. During hypoxia, the PMN metabolic switch towards glycolysis is required for mitochondrial ROS production, and this is essential for HIF-1α stabilization [84] and the release of NETs. In human neutrophils, a correlation between mTOR-dependent upregulation of HIF-1α and NETs formation has been demonstrated after LPS stimulation [85], but excess NET formation and therefore excessive inflammation can damage the host. For example, during sepsis, neutrophil dysfunction has been reported, and the inhibition of glycolysis via downregulation of the PI3K/Akt-HIF-1α pathway seems to contribute to neutrophil immunosuppression [86]. Under hypoxic conditions, HIF-1 triggers NF-κB in PMNs to promote the survival of these cells. Increased levels of NF-κB encourage neutrophil activation, which increases the generation of nitric oxide and pro-inflammatory cytokines while decreasing cellular apoptosis [87].

Besides HIF-1α, HIF-2 seems to be involved in the regulation of PMNs, and, in particular, its expression is increased in the context of inflammatory diseases [88]. 

In terms of cytotoxicity, natural killer (NK) lymphocytes are the major innate effector cells, especially against tumor [89], and the tumor microenvironment is often characterized by hypoxia, which promotes cell proliferation and angiogenesis [90]. NK cells are characterized by the presence on their surface of activating and inhibitory receptors, and a balance between the two is essential for the recognition of target cells [91]. When exposed to low-oxygen environments, NK cells show impaired cytolytic function [92], increased expression and stabilization of HIF-1α, and major responsiveness to IL-2, which mediates NK cell proliferation, in order to execute their functions [93]. Upregulation of HIF-1α in NK cells requires both hypoxia and IL-2 stimulation, since hypoxia is implicated in HIF-1α stabilization, whereas IL-2 leads to the activation of PI3K/mTOR signaling, which is vital for HIF-1α protein synthesis [94]. The balance between activating and inhibitory receptors might also be negatively affected by hypoxia. This condition leads to an impaired transcription of some of the receptors implicated in NK cell activation, such as NKp46, NKp30, NKp44, and NKG2D, resulting in impaired capacity to kill infected cells or tumor cells [95]. NK cells also undergo metabolic reprogramming when exposed to hypoxia in the tumor microenvironment. In patients with liver cancer, NK cells were found with fragmented mitochondria, which causes less oxidative phosphorylation and therefore low ATP production; moreover, they exhibit lower expression of granzyme B. Hypoxic conditions upregulate Drp1, a central player in mitochondrial fission. Zheng et al., highlighted how a low oxygen environment results in constant activation of mTOR-Drp1 GTPase in NK cells in tumor tissue, causing excessive mitochondrial fission into fragments. All of these features are related to reduced cytotoxicity and NK cell loss, which promote tumor evasion of NK cell-mediated surveillance [96]. A decline in adaptive immune responses occurs with the aging process, leading to increased susceptibility to infection due to thymic atrophy, a reduction in the number of peripheral blood naïve cells, and an increase in memory cells [97]. In older people, T-cell mitochondria are characterized by impaired oxidative phosphorylation even though they contain a larger quantity of protein than younger subjects. This impaired function causes a dysregulation of aged T-cell signaling pathways and leads to the activation of the inflammasome together with an aged T-cell proinflammatory phenotype through the activation of PI3K-AKT-mTOR and MAPK signaling [98]. Upregulation of glycolysis is mandatory for effector T-cell differentiation, and the activation of HIF-1α through metabolic reprogramming towards glycolysis can improve effector cell functions. Inhibition of glycolysis results in T-cell anergy and in a shift of effector T cells towards Tregs, whose metabolism is based on lipid oxidation and oxidative phosphorylation [99]. The development and differentiation of T lymphocytes is significantly influenced by the transcription factor NF-κB. It participates in the development of T cells from the early stage of thymocyte differentiation to post-selection maturation. It has been proven that there is a cross-talk between HIF signals and NF-κB [100], and according to Bruzzese et al., NF-κB activation increases the sensitivity of T cells to hypoxia. Moreover, both signals are important in the differentiation of regulatory T cells [101].

HIF-1α has also a key role in B-cell metabolism and functions. B-cell development and differentiation is affected by the different levels of oxygen to which they are exposed, especially in the light zones of the germinal centers that are known to be hypoxic. HIF-1α plays a vital role in hypoxia-induced B-cell cycle arrest and causes an increase in glycolytic metabolism while reducing B-cell proliferation and increasing B-cell death [102]. In germinal center B cells, HIF-1 is highly expressed. High-affinity plasma cells were produced as a result of defective class-switch recombination caused by impaired germinal center in response to HIF-1α knockout B cells [103].

A cross-talk between HIF, aging, and immune cells is undoubtedly present, and an implication of HIF dysregulation is expected in many different immune-mediated diseases, including sepsis, cancer, and inflammatory bowel disease [104]. Therefore, more studies on this topic can lead us to understand the underlying mechanisms of aging in immune cells when exposed to a hypoxic environment. (Figure 3).

## 5. Hypoxia, Aging, and Biomarkers

Hypoxia-inducible factor-1 is involved in recruitment of innate and adaptive immune cells in the vascular system, causing vascular inflammation and the release of cytokines. Following the interactions between pro-inflammatory cytokines and vascular smooth muscle cells (VSMCs), oxidative stress occurs, resulting in vascular remodeling and hypertension, which have a key mechanistic role in the development of cardiovascular aging-related disease [105].

It has been described that angiotensin II could increase monocyte chemoattractant protein 1 (MCP-1) expression, which plays a pivotal role in the regulation of the migration and infiltration of monocytes within injured tissue [106]. Moreover, angiotensin II treatment is associated with increased proinflammatory cytokine production, such as IL-1b, IL-6, and TNFα, which drives vascular remodeling as a consequence of vascular inflammation [107].

A study by Qi et al., on the role of HIF-1α signaling in VSMCs showed that inhibition of HIF-1α or its downstream CCL7 (chemokine C-C motif ligand 7), a chemokine involved in monocyte recruitment, can reduce angiotensin II-related macrophage infiltration. This can therefore be a potential therapeutic strategy to treat vascular remodeling-related diseases [108].

Different age-related biomarkers have been evaluated in different cells under hypoxic conditions (Table 1). β-galactosidase activity, p16 gene expression, and proliferation rate are considered classic markers of senescence in cellular biology. β-galactosidase is a lysosomal hydrolase whose activity is increased due to increased autophagy, which can be linked to augmented lysosomal content in senescent cells [109]. p16 is a tumor suppressor gene involved in aging that leads to G1 cell cycle arrest through the retinoblastoma (Rb) pathway and inhibition of the action of cyclin-dependent kinases [110]. Damiani et al., showed a correlation between low atmospheric oxygen tension (5% O_2_) exposure in human embryonic diploid fibroblasts and a reduced expression of β-galactosidase activity and p16 compared to human fibroblasts exposed to 21% O_2_. Moreover, cells cultured under 5% O_2_ showed a slower proliferation rate and therefore a slowdown in the aging process. They showed that serial passaging of cells under normoxia and hypoxia leads to cell aging [111]. The use of biomarkers in the aging process is an emerging field of research [112]. Moaddel et al., identified 232 proteins involved in the molecular mechanisms of aging and described 21 signaling pathway groups that underlie biological changes of senescent cells, such as IGF1, advanced glycation end-products (AGEs) and advanced glycation end-product receptors (RAGEs), FOXO, and signaling cytokine pathways. The expression of human IGF1 in senescent cells is reduced, and higher levels are associated with lower mortality. AGEs are proteins that increase in concentration with aging and when bound to RAGEs trigger several signaling pathways, such as HIF-1 and MAPK, which are related to the aging process. Forkhead box O3 (FOXO) is a family of transcription age-related factors that control the proteins involved in energy homeostasis, ROS detoxification, superoxide dismutase 2 (SOD2), and epidermal growth factor receptor-signaling pathway [113].

## 6. Conclusions

Human organisms have developed a sensing system that rapidly responds to changes in tissue oxygenation through the induction of HIFs in order to ensure production of ATP and therefore cellular survival. During the aging process, several signaling path-ways are altered, including those involved in immunosenescence. The activation of HIF pathways is involved in adaptive processes when immune cells are exposed to hypoxia, but in this context, other studies are needed to improve our knowledge about the mechanisms that underlie the responses to hypoxia and possibly consider these pathways for the development of future therapeutic strategies.

## Figures and Tables

**Figure 1 biomedicines-11-02163-f001:**
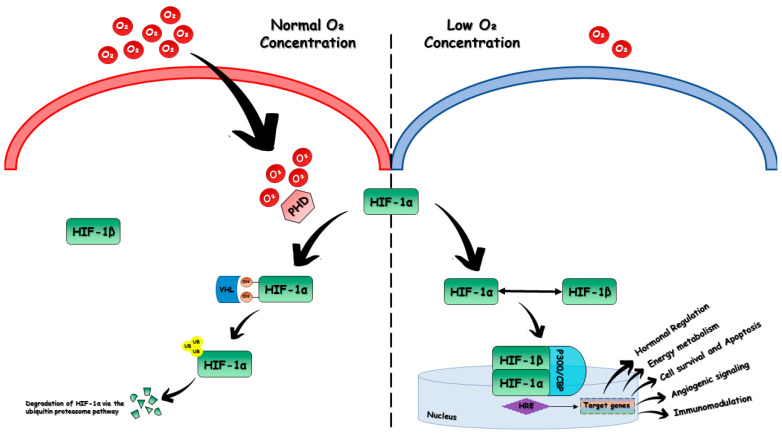
**Mechanism of oxygen-dependent regulation of HIF-1α.** In normoxic conditions, HIF-1β is expressed constitutively, whereas HIF-1α is hydroxylated in its proline residues by prolyl hydroxylase proteins (PHDs), an oxygen-sensing system that uses oxygen as a substrate. This allows for its degradation, mediated by the von Hippel–Lindau tumor suppressor protein (VHL) via the ubiquitin proteasome pathway. Under low oxygen concentration, the VHL tumor suppressor protein becomes inactivated, which allows HIF-1α and HIF-1β to form a heterodimer in the nucleus, which, by binding to p300/CBP, forms a transcriptional activation complex that binds to the hypoxia response element (HRE) and functions as a transcription factor.

**Figure 2 biomedicines-11-02163-f002:**
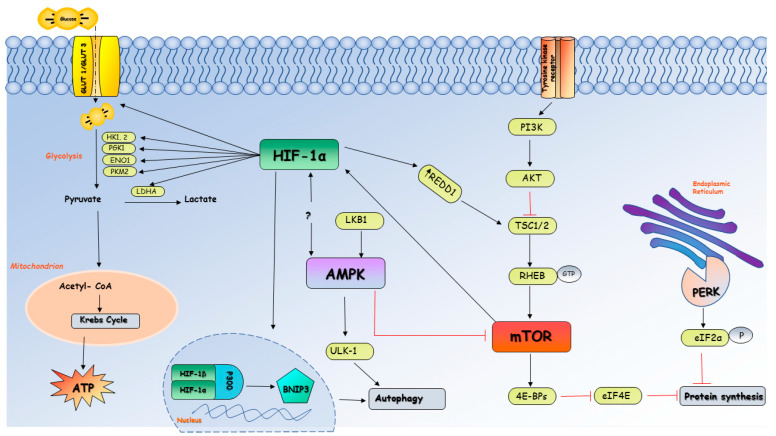
**Metabolism of cells exposed to hypoxia and the role of HIF-1α.** Under low oxygen conditions the cell metabolism is reprogrammed by HIF-1α through a reduction in cellular oxygen dependence, which allows cell survival and maintains host homeostasis. HIF-1α increases the transcription of several genes that encode for proteins involved in the glycolytic pathway, such as GLUT1 and GLUT3, hexokinases (HK1, 2), phosphoglycerate kinase (PGK1), enolase (ENO1), pyruvate kinase (PKM1), and lactate dehydrogenase A (LDHA), which are useful for the production of ATP in hypoxic cells. A reduction in ATP:AMP ratio in cells exposed to low oxygen concentrations leads to the activation of AMPK through the LKB1-AMPK axis, causing an inhibition of mTOR and therefore protein synthesis. Downregulation in protein synthesis occurs through two major pathways, mTOR (mammalian target of rapamycin) complex and PERK (protein kinase R-like ER kinase), in order to suppress cell expensive energy processes. Hypoxia negatively regulates mTOR-related protein translation, which is no longer able to phosphorylate eIF4E-binding proteins (4E-BPs) and therefore prevents the assimilation of eIF4E into a complex necessary for binding to mRNA cap and initiates translation. Moreover, PERK phosphorylates eukaryotic initiation factor 2 alpha (eIF2α) and inhibits translation initiation to prevent abnormal accumulation of unfolded/misfolded proteins that can cause endoplasmic reticulum stress. To optimize the use of limited energy supplies, the cell activates self-preservation process such as autophagy through HIF-1α-dependent expression of BNIP3 (Bcl-2/adenovirus E1B 19-kDa-interacting protein) and HIF-independent mechanism ULK1-AMPK.

**Figure 3 biomedicines-11-02163-f003:**
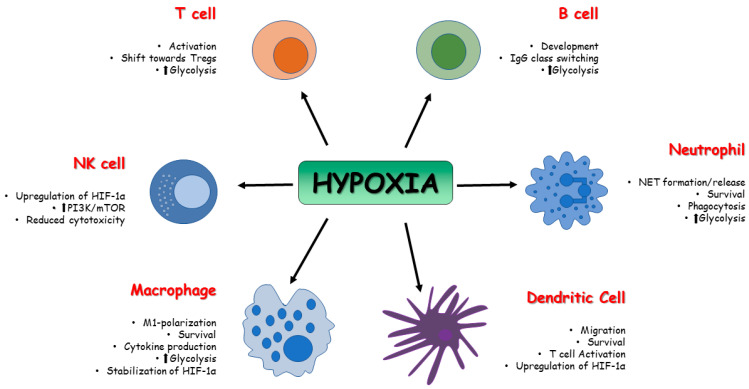
**Effect of HIFs on immune cell metabolism**. The activity of HIFs affects the function and the metabolism of innate and adaptive immune cells in terms of survival, activation, development, and polarization.

**Table 1 biomedicines-11-02163-t001:** Age-related biomarkers and their expression in hypoxic cells.

Biomarker	Function	In Hypoxic Cells
β-galactosidase	Lysosomal hydrolase	Reduced expression
p16	Tumor suppressor gene	Reduced expression
CCL7 (chemokine C-C motif ligand 7)	Chemokine involved in monocyte recruitment	Increased expression
IGF1 (insulin-like growth factor-1)	Growth factor	Increased expression
AGEs (advanced glycation end- products)	Molecules involved in vascular complications	Increased expression
RAGEs (advanced glycation end-product receptors)	Molecules involved in vascular complications	Increased expression
FOXO (Forkhead box O3)	Transcription factors	Increased expression

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
