# Peer review of "Hypoxic State of Cells and Immunosenescence: A Focus on the Role of the HIF Signaling Pathway"

_biomedicines, 2023, doi:10.3390/biomedicines11082163_

Round 1

Reviewer 1 Report

The review by Troise D. et al attempts to elucidate the mechanisms of aging by studying events associated with hypoxic state of cells, and focused on the role of the HIF signaling pathway. The review is interesting and covers in sufficient detail the overall characteristics of HIF factors, its role in cell metabolic status in hypoxia and effect on the immune cells.

At the same time there are some remarks to the authors. The abstract rather resembles an introduction and does not clearly reflect the content of the review. As the scope of the review is limited by HIF pathway, the effect of HIF asparaginyl hydroxylase should be described in more details and may be included in left part of Fig.1which logically must contain also a free form of HIF-1β as this factor is expressed constitutively.

Also a part (lines 286-299) describing ischemic preconditioning which is widely known as a protect factor against IR damage should be increased. The participation of HIF pathway especially in case of heart IR is proven.

There are also some technical shortcomings. Some phrases, for example “Oxygen (O2) has a crucial role in cell metabolism, it is both a substrate and a cofactor for many enzymes”, were moved from one part of the text to another part without any change. In line 324 the abbreviation DC(Dendritic cells) was transformed to CD.

Nevertheless the review is useful and can be published after correction.

Author Response

We thank the reviewer for these suggestions. Please find  a point-by point answer to the specific comments

  • Comment 1: The abstract rather resembles an introduction and does not clearly reflect the content of the review.

Response: Thank you for pointing this out. We have revised the abstract and organized it better so that it provides a summary of the review's most important points. (Lines 14-22)

  • Comment 2: As the scope of the review is limited by HIF pathway, the effect of HIF asparaginyl hydroxylase should be described in more details and may be included in left part of Fig.1which logically must contain also a free form of HIF-1β as this factor is expressed constitutively.

Response: We have modified the Fig.1 and further described the effect of HIF asparaginyl hydroxylase. (Lines 131-147)

  • Comment 3: Also a part (lines 286-299) describing ischemic preconditioning which is widely known as a protect factor against IR damage should be increased. The participation of HIF pathway especially in case of heart IR is proven.

Response: We agree with this suggestion and we have integrated this recommendation into the manuscript (Lines 308-316)

  • Comment 4: Some phrases, for example “Oxygen (O2) has a crucial role in cell metabolism, it is both a substrate and a cofactor for many enzymes”, were moved from one part of the text to another part without any change. In line 324 the abbreviation DC(Dendritic cells) was transformed to CD.

Response: We have fixed the typo and the repetitions into the manuscript.

Reviewer 2 Report

Title: Hypoxia Inducible Factors and limited energy supplies in cellular aging

Authors: Troise Dario, Infante Barbara, Mercuri Silvia, Netti Giuseppe Stefano, Ranieri Elena, Gesualdo Loreto, Stallone Giovanni and Pontrelli Paola

Summary:

In this review, the authors focused on the mechanisms underlying the aging process after events leading to a state of cellular hypoxia, with emphasis on the HIF signaling pathway. This manuscript is an interesting work, however, the missing information needs to be filled in and if it is not available, the limitations of the available results need to be discussed in detail.

Several points are listed below:

1: The title is confusing and does not say what the authors actually want to say. It should be fundamentally rewritten!

2: The abstract should be rewritten as it contains many repetitions and appears unstructured.

3: The novelty of the article should be emphasized more clearly throughout, which is not the case here.

4: The introduction should be restructured from scratch.

5: Limitations related to HIF signaling pathway and aging should be discussed in more detail.

6: I miss in this paper a systematic table on hypoxia, aging and biomarkers.

7: The authors claim in conclusion that "all of these age-related changes lead to increased cellular damage, resulting in damage throughout the body, and hypoxia may contribute to this damage." What do they mean in great detail and should be discussed more.

8: All abbreviations should be defined at their first mention and used thereafter.

9: The paper contains many complicated abbreviations. A list of the most important abbreviations would be helpful to the reader.

10: The English language in general needs improvement. The entire text needs to be revised by a native English speaker.

Moderate editing of English language required.

Author Response

We thank the reviewer for the powerfull suggestions. Please find a poin-by-point answer to specific comments

  • Comment 1: The title is confusing and does not say what the authors actually want to say. It should be fundamentally rewritten!

Response: Thank you for making this observation. According to that, the title has been rewritten to provides a description of the content of the article.

  • Comment 2: The abstract should be rewritten as it contains many repetitions and appears unstructured.

Response: Thank you for pointing this out. We have revised it and organized it better so that it provides a summary of the review's most important points. (Lines 14-22)

  • Comment 3: The novelty of the article should be emphasized more clearly throughout, which is not the case here.

Response: We agree with the reviewer that the original manuscript needed improvement to emphasized its novelty but thanks to the reviewer’s suggestions now the article focus more on the hypoxic state of cells and immunosenescence than before.

  • Comment 4: The introduction should be restructured from scratch.

Response: We have streamlined and written again the introduction to give the reader the focus on the topic of the review. (Lines 44-63)

  • Comment 5: Limitations related to HIF signaling pathway and aging should be discussed in more detail.

Response: Thank you for making the observation. In the conclusions we highlighted the need of more in-depth studies to improve our knowledge about the mechanisms that underlie the responses to hypoxia and possibly consider these pathways for the development of future therapeutic strategies (lines 500-507)

  • Comment 6: I miss in this paper a systematic table on hypoxia, aging and biomarkers.

Response: We agree with this and included the suggested information in Table 1 .

  • Comment 7: The authors claim in conclusion that "all of these age-related changes lead to increased cellular damage, resulting in damage throughout the body, and hypoxia may contribute to this damage." What do they mean in great detail and should be discussed more.

Response: We regret that the results were not clear. We proceeded to rewrite the conclusions to make them clearer. (lines 500-507)

  • Comment 8: All abbreviations should be defined at their first mention and used thereafter.

Response: Thank you for making this observation. We have revised the manuscript to define all abbreviations at the first mention.

  • Comment 9: The paper contains many complicated abbreviations. A list of the most important abbreviations would be helpful to the reader.

Response: We added a list of abbreviations to the manuscript.

  • Comment 10: The English language in general needs improvement. The entire text needs to be revised by a native English speaker.

Response: Thank you for making this observation. We had the paper reviewed by an English speaker.

Reviewer 3 Report

The manuscript attempts to summarize the function and activity of HIF proteins in the context of diseases and aging. The authors discuss the mechanisms through which HIF proteins impact cells of innate and adaptive systems with aging.

Comments: Overall, the scope of the review is broad, with little focus on the role of HIFs in the molecular mechanisms of aging. As it stands, the title of the manuscript is not appropriate. Excessive use of review citations (up to 65% of cited papers are review papers) does not help to convey a clear message.

Abstract. The following sentence doesn't recapitulate what is discussed or highlighted in the review "Despite its complex role in the regulation of the inflammatory responses, hypoxia seems to exert a net anti-inflammatory effect through HIF activation." It only adds confusion to the message, as systemic markers of inflammation increase with age.

Authors should rewrite and shorten the introduction to present the essence of the review in a synthesized form. In its current form, the introduction summarizes the reviews' contents in a form that resembles a bullet point presentation.

Page 3 of 15; lines 101-122: The first three paragraphs contain useless general information. I would delete those from the manuscript.

The authors should consider reorganizing several sections to make reading the manuscript easier. Eliminating non-essential details that distract us more than anything else could offer a more compelling thread to the review.

Although a few sentences could be more concise, the English style/grammar is correct.

Author Response

We thank the reviewer for the powerfull suggestions. Please find a poin-by-point answer to specific comments

  • Comment 1: Overall, the scope of the review is broad, with little focus on the role of HIFs in the molecular mechanisms of aging. As it stands, the title of the manuscript is not appropriate. Excessive use of review citations (up to 65% of cited papers are review papers) does not help to convey a clear message.

Response: Thank you for making this observation. We expanded bibliography and rewrite the title to be more appropriate.

  • Comment 2: The following sentence doesn't recapitulate what is discussed or highlighted in the review "Despite its complex role in the regulation of the inflammatory responses, hypoxia seems to exert a net anti-inflammatory effect through HIF activation." It only adds confusion to the message, as systemic markers of inflammation increase with age.

Response: We have revised the abstract and organized it better to convey a clearer message. (Lines 14-22)

  • Comment 3: Authors should rewrite and shorten the introduction to present the essence of the review in a synthesized form. In its current form, the introduction summarizes the reviews' contents in a form that resembles a bullet point presentation.

Response: We have streamlined the introduction by eliminating any irrelevant information to give the reader the preliminary information to understand the paper. (Lines 43-63)

  • Comment 4: Page 3 of 15; lines 101-122: The first three paragraphs contain useless general information. I would delete those from the manuscript.

Response: We agree and deleted them.

  • Comment 5: The authors should consider reorganizing several sections to make reading the manuscript easier. Eliminating non-essential details that distract us more than anything else could offer a more compelling thread to the review.

Response: We appreciate the reviewer’s insightful suggestion and tried to reduce some sections to make the manuscript easier to read.

Reviewer 4 Report

In this review, the authors discuss the implication of hypoxia, HIFs, energy status, and aging. While this is an interesting topic, there are many points that contain errors (especially on HIFs and hypoxia)  that can cause misunderstanding (especially to the non-expert) that must be corrected. Furthermore, the text needs meticulous language correction and proofreading. As implied by its title, this is a review of hypoxia, aging but also cellular energy status. Although in the first part, the authors discuss the Hypoxia/HIFs axis and energy metabolism (based on the AMPK pathway), the latter and, possibly, the most interesting part deals only with hypoxia immune response and aging but lacks information on their possible interaction on both pathways previously discussed. Furthermore, the aging part of this review, which is supposed to be the subject, is extremely limited in the last section.

Specific points

1) As discussed in the previous paragraph, the main point of criticism is that this review needs an expansion on information about aging. Also, the interplay and the possible connection between the two main pathways discussed in the first paragraphs should be given in order to give context to this review.

2) The paragraph between lines 146-153 contains significant errors on HIFs and their pathway. HIF-1,-2, and 3α are not splicing variants! (Lines 146-147). They are products of different genes (HIF1A, EPAS1, HIF3A). Although, indeed, HIF-1 and HIF-2 have different specificities, the mention that mainly HIF-2 mediates lipid metabolism (line 151) is not correct as there are many recent reports that HIF-1 also mediates the expression of genes that produce significant enzymes of lipid metabolism (e.g. FASN, LPIN1, AGPAT2….). Finally, in the same paragraph concerning HIF-3α (that possesses a few splicing variants), it can act as a negative regulator of HIFs but also potentiate transcription by itself depending on the HIF-3α isoform (Tolonen JP et al CMLS 2020). Most important, the HIF-3alpha splicing variant that acts negatively on HIF-signaling binds to ARNT and not to HIF-1α or HIF-2α as the authors state.

3) As the immune response is many times stated in the manuscript, the authors completely ignore recent findings on HIF-1 and NF-kB interplay. Furthermore, concerning immune response and cancer, HIF-1 mediates the expression of PD-L1, which contributes to immune suppression. 

4) As the authors discuss the effect of HIFs on energy metabolism, their effects on mitochondrial processing should be further discussed. For example, HIF-1 not only regulates acetyl-CoA flux to the Krebs cycle by inducing PDK1 that regulates PDH (Lee P, Nat Rev Mol Cell Biol. 2020) but also mediates the expression of different COX isoforms to regulate respiration (Fukuda Cell 2007) e.t.c.

5) Although authors try to implicate AMPK and HIF pathways, they do not discuss that in LKB1 or AMPK deficient cells, HIF-1α is overexpressed.

6) Lines 234-235 AKT/mTOR pathway promotes HIF-α translation, not transcription.

7) Minor but important. It is not only one PHD enzyme but more. So, a plural must be used.

8) Finally, there are numerous grammatical and linguistic errors that need correction. E.g. wrong choice of verbs (exposition is not synonym to exposure), spelling errors (HIF, not HOF), and many others.

The text needs meticulous language correction and proofreading.  There are numerous grammatical and linguistic errors that need correction. E.g. wrong choice of verbs (exposition is not synonym to exposure), spelling errors (HIF, not HOF), and many others.

Author Response

We thank the reviewer for the powerfull suggestions. Please find a poin-by-point answer to specific comments

  • Comment 1: As discussed in the previous paragraph, the main point of criticism is that this review needs an expansion on information about aging. Also, the interplay and the possible connection between the two main pathways discussed in the first paragraphs should be given in order to give context to this review.

Response: The manuscript was revised and now focus more than before on the hypoxic state of cells and immunosenescence .

  • Comment 2: The paragraph between lines 146-153 contains significant errors on HIFs and their pathway. HIF-1,-2, and 3α are not splicing variants! (Lines 146-147). They are products of different genes (HIF1A, EPAS1, HIF3A). Although, indeed, HIF-1 and HIF-2 have different specificities, the mention that mainly HIF-2 mediates lipid metabolism (line 151) is not correct as there are many recent reports that HIF-1 also mediates the expression of genes that produce significant enzymes of lipid metabolism (e.g. FASN, LPIN1, AGPAT2….). Finally, in the same paragraph concerning HIF-3α (that possesses a few splicing variants), it can act as a negative regulator of HIFs but also potentiate transcription by itself depending on the HIF-3α isoform (Tolonen JP et al CMLS 2020). Most important, the HIF-3alpha splicing variant that acts negatively on HIF-signaling binds to ARNT and not to HIF-1α or HIF-2α as the authors state.

Response: We thank the reviewer for this comment. We have revised the text to address these points.  (Lines 84-104)

  • Comment 3: As the immune response is many times stated in the manuscript, the authors completely ignore recent findings on HIF-1 and NF-kB interplay. Furthermore, concerning immune response and cancer, HIF-1 mediates the expression of PD-L1, which contributes to immune suppression.

Response: Thank you for pointing this out. We've included these findings in the text. (Lines 321-331 – 354-359 – 393-396 – 432-437.)

  • Comment 4: As the authors discuss the effect of HIFs on energy metabolism, their effects on mitochondrial processing should be further discussed. For example, HIF-1 not only regulates acetyl-CoA flux to the Krebs cycle by inducing PDK1 that regulates PDH (Lee P, Nat Rev Mol Cell Biol. 2020) but also mediates the expression of different COX isoforms to regulate respiration (Fukuda Cell 2007) e.t.c.

Response: We thank the reviewer for these suggestions to improve the quality of the manuscript. In the novel version of the review we discussed all these aspects raised by the reviewer. (Lines 171-195)

  • Comment 5: Although authors try to implicate AMPK and HIF pathways, they do not discuss that in LKB1 or AMPK deficient cells, HIF-1α is overexpressed.

Response: We have revised the manuscript to discuss this important point (Lines 207-219)

  • Comment 6: Lines 234-235 AKT/mTOR pathway promotes HIF-α translation, not transcription.

Response: We have fixed the error into the manuscript.

  • Comment 7: Minor but important. It is not only one PHD enzyme but more. So, a plural must be used.

Response: Thank you for the comment, we corrected the text accordingly.

  • Comment 8: Finally, there are numerous grammatical and linguistic errors that need correction. E.g. wrong choice of verbs (exposition is not synonym to exposure), spelling errors (HIF, not HOF), and many others.

Response: We appreciate your thoughtful response and fixed the grammatical and linguistic errors.

Round 2

Reviewer 2 Report

The authors have addressed all points of potential criticism and each suggestion made by the reviewer adequately and in detail.

Minor editing of English language required.

Author Response

We thank the reviewer for this endorsement. In the revised version of the manuscript minor editing of English language has been provided.

Reviewer 3 Report

The authors have strengthened the manuscript. Several criticisms were addressed correctly. Overall, I am satisfied with the responses and changes made by the authors.

The manuscript remains wordy, with long and complex sentences. Grammatical corrections are still necessary. 

Proofreading by a native English speaker. 

Author Response

We thank the reviewer for this endorsement. In the revised version of the manuscript we revised long sentences and provided editing of English language.

Reviewer 4 Report

The authors have tried and significantly improved the manuscript. However, before being accepted there must be some minor corrections.

1) Lines 77-79: The sentence "CTAD (carboxyl-terminal domain), that regulates the transcription of genes under low level oxygen conditions and NTAD (amino-terminal domain), implicated in stabilization of HIF-1α". CTAD and NTAD stand for Carboxy- or amino- terminal Transctivation Domains. Furthermore, NTAD is implicated in the selection or target genes and not in stability (as the authors mention). It just resides inside the ODDD domain (which is responsible for stability).

2) There is an issue with the uniformity of symbols used for the same molecule to avoid confusion. e.g. the authors use HIF-1 or HIF1 (choose one ; HIF-1 is the main version), at the same line 109 proline is symbolized both PRO and pro). 

3) Please proofread, there are still text that plural is used instead on singular. e.g. line 130 it must be FIH suppresses; line 308 "cell death was reduced" etc. there are other places inside the text that need this kind of corrections.

It has been improved however there are sections that need grammar checks.

Author Response

We thank the reviewer for this endorsement. In the revised version of the manuscript we provided editing of English language. We also considered the other specific suggestions by this reviewer and made corrections to the manuscript (Lines 76-79, line 109, line 130, line 308).